# Real-World Diagnostic Accuracy of the On-Site Cytopathology Advance Report (OSCAR) Procedure Performed in a Multidisciplinary One-Stop Breast Clinic

**DOI:** 10.3390/cancers15204967

**Published:** 2023-10-13

**Authors:** Voichita Suciu, Carolla El Chamieh, Ranya Soufan, Marie-Christine Mathieu, Corinne Balleyguier, Suzette Delaloge, Zsofia Balogh, Jean-Yves Scoazec, Sylvie Chevret, Philippe Vielh

**Affiliations:** 1Gustave Roussy, Université Paris-Saclay, 94805 Villejuif, France; 2Department of Biostatistics and Medical Information, INSERM UMR1153 ECSTRRA Team, Hôpital Saint Louis, AP-HP, 75010 Paris, France; 3Medipath and American Hospital of Paris, 92200 Paris, France

**Keywords:** breast mass, fine-needle aspiration (FNA), cytomorphology, diagnostic accuracy, one-stop clinic, interventional cytopathologist

## Abstract

**Simple Summary:**

The aim of our study was to assess the diagnostic accuracy of the on-site cytopathology advance report (OSCAR) procedure on breast FNA cytologic samples in our breast OSC during the first three years (April 2004 till March 2007) of its implementation. To this goal, we retrospectively analyzed a series of 1820 breast masses (1740 patients) radiologically classified according to the American College of Radiology (ACR) BI-RADS lexicon (67.6% being either BI-RADS 4 or 5), sampled by FNA and immediately diagnosed by cytomorphology. The on-site cytopathology advance report (OSCAR) procedure is a highly reliable diagnostic approach for breast masses, provided that it is performed by interventional cytopathologists in the multidisciplinary setting of a one-stop clinic (OSC). Besides dramatically limiting the rate of unsatisfactory specimens and diagnostic turnaround time, OSCAR is a highly efficient first-line approach. Diagnostic accuracy measures of the OSCAR procedure with their 95% confidence intervals (95% CI) were the following: sensitivity (Se) = 97.4% (96.19–98.31); specificity (Sp) = 94.98% (92.94–96.56); positive predictive value (PPV) = 96.80% (95.48–97.81); negative predictive value (NPV) = 95.91% (94.02–97.33); positive likelihood ratio (LR+) = 19.39 (13.75–27.32); negative predictive ratio (LR−) = 0.03 (0.02–0.04), and; accuracy = 96.45% (95.42–97.31). The respective positive likelihood ratio (LR+) for each of the four categories of cytopathological diagnoses (with their 95% CI) which are malignant, suspicious, benign, and nondiagnostic were 540 (76–3827); 2.69 (1.8–3.96); 0.03 (0.02–0.04); and 0.37 (0.2–0.66), respectively.

**Abstract:**

Fine-needle aspiration (FNA) cytology has been widely used for the diagnosis of breast cancer lesions with the objective of differentiating benign from malignant masses. However, the occurrence of unsatisfactory samples and false-negative rates remains a matter of concern. Major improvements have been made thanks to the implementation of rapid on-site evaluation (ROSE) in multidisciplinary and integrated medical settings such as one-stop clinics (OSCs). In these settings, clinical and radiological examinations are combined with a morphological study performed by interventional pathologists. The aim of our study was to assess the diagnostic accuracy of the on-site cytopathology advance report (OSCAR) procedure on breast FNA cytologic samples in our breast OSC during the first three years (April 2004 till March 2007) of its implementation. To this goal, we retrospectively analyzed a series of 1820 breast masses (1740 patients) radiologically classified according to the American College of Radiology (ACR) BI-RADS lexicon (67.6% being either BI-RADS 4 or 5), sampled by FNA and immediately diagnosed by cytomorphology. The clinicoradiological, cytomorphological, and histological characteristics of all consecutive patients were retrieved from the hospital computerized medical records prospectively registered in the central information system. Histopathological analysis and ultrasound (US) follow-up (FU) were the reference diagnostic tests of the study design. In brief, we carried out either a histopathological verification or an 18-month US evaluation when a benign cytology was concordant with the components of the triple test. Overall, histology was available for 1138 masses, whereas 491 masses were analyzed at the 18-month US-FU. FNA specimens were morphologically nondiagnostic in 3.1%, false negatives were observed in 1.5%, and there was only one false positive (0.06%). The breast cancer prevalence was 62%. Diagnostic accuracy measures of the OSCAR procedure with their 95% confidence intervals (95% CI) were the following: sensitivity (Se) = 97.4% (96.19–98.31); specificity (Sp) = 94.98% (92.94–96.56); positive predictive value (PPV) = 96.80% (95.48–97.81); negative predictive value (NPV) = 95.91% (94.02–97.33); positive likelihood ratio (LR+) = 19.39 (13.75–27.32); negative predictive ratio (LR−) = 0.03 (0.02–0.04), and; accuracy = 96.45% (95.42–97.31). The respective positive likelihood ratio (LR+) for each of the four categories of cytopathological diagnoses (with their 95% CI) which are malignant, suspicious, benign, and nondiagnostic were 540 (76–3827); 2.69 (1.8–3.96); 0.03 (0.02–0.04); and 0.37 (0.2–0.66), respectively. In conclusion, our study demonstrates that the OSCAR procedure is a highly reliable diagnostic approach and a perfect test to select patients requiring core-needle biopsy (CNB) when performed by interventional cytopathologists in a multidisciplinary and integrated OSC setting. Besides drastically limiting the rate of nondiagnostic specimens and diagnostic turn-around time, OSCAR is an efficient and powerful first-line diagnostic approach for patient-centered care.

## 1. Introduction

Fine-needle aspiration (FNA) cytology of breast masses is a useful tool for the rapid differential diagnosis of breast masses. Its advantages are the absence of medical contraindications, the fact that it is minimally invasive, safe, reliable, cost-effective, and reduces turn-around time considerably, thus mitigating patient anxiety in anticipation of diagnostic results [1,2,3,4,5,6,7,8,9,10,11,12,13,14,15,16,17,18,19,20,21,22,23,24,25,26,27,28,29,30,31,32,33,34,35,36,37,38,39,40,41,42,43,44,45,46,47,48,49,50,51,52,53,54,55,56,57,58,59,60,61,62,63,64,65,66,67,68,69,70,71,72,73,74,75,76,77,78,79,80,81,82,83,84,85,86,87,88,89,90,91,92,93,94,95,96,97,98,99,100,101,102,103,104,105]. 

Despite its many advantages, variable nondiagnostic rates and the occurrence of false-negative and false-positive results, mainly operator-dependent, remain a matter of concern. Owing to these limitations, the debate over the current place of FNA as compared to core-needle biopsy (CNB) still persists [106,107,108,109,110]. CNB is now mandatory for the pre-operative diagnosis of patients with breast cancer for tumor grading and immunohistochemical evaluation of the prognostic biomarkers as well as other markers that could influence the primary treatment choice.

In view of the non-invasive nature of FNA, its quickness and low cost coupled with the high sensitivity and specificity rates in the diagnosis of breast masses, major efforts have been dedicated to improving its overall diagnostic performance in pathology practice. In this context, rapid on-site evaluation (ROSE) has represented a key game changer, particularly in OSCs in which reducing turnaround time and mitigating patient anxiety are two major priorities.

Due to the time of processing, the use of CNBs only for patients with breast masses may delay the diagnosis if FNA is not immediately used. By contrast, in the setting of an OSC, FNA combined with ultrasound (US) guidance is very suitable for accurate and same-day diagnosis of breast masses. FNA not only minimizes the time to diagnosis but also significantly decreases unsatisfactory specimen rates. This has been made possible thanks to the application of ROSEs to FNA samples [111,112]. Studies show that having an interventional pathologist make an on-the-spot cytological evaluation is a crucial part of the highly efficient triple test [113] and may sharply reduce unnecessary CNBs and surgery for benign lesions [93]. Our OSC for breast masses was opened in 2004 and is located in the outpatient clinic of the tertiary referral Gustave Roussy (GR) comprehensive cancer center (Villejuif, France). Data about the challenges and cost-effectiveness have been previously published [114].

The goal of our retrospective study was first to assess the diagnostic accuracy of the OSCAR procedure in a large consecutive series of patients with breast masses examined during the first 3 years of our OSC. To this aim, we compared cytopathological diagnoses with corresponding histopathological examinations. Second, in cases showing concordant benign clinicoradiological results and cytopathological diagnoses, we examined the US characteristics of the breast masses after 18 months with a US follow-up (US-FU). Finally, we evaluated the rate of nondiagnostic specimens sampled by FNA and described the features of the breast masses whose cytomorphology were discordant with the histopathological findings.

## 2. Patients and Methods

A retrospective observational study was conducted in a consecutive series of 1820 breast masses from 1740 patients sampled by FNA. The analysis covered our first 3 y experience (5 April 2004 till 31 March 2007) at an OSC set up at the GR institution—a large cancer comprehensive center distinct from regional screening facilities. 

### 2.1. Organization of the Breast One-Stop Clinic (OSC)

The objective of the OSC was to identify patients with asymptomatic (non-palpable) breast masses detected either by the French national breast cancer screening program launched in 2004 or through opportunistic screening or diagnostic mammography in symptomatic patients with palpable lumps. Patients were evaluated in a large-scale multidisciplinary outpatient breast clinic. Patients with pure microcalcifications without a breast mass detected by mammography were not sampled by FNA. 

The requirements for setting up a dedicated breast cancer OSC [115,116] have already been described. In brief, a dedicated hotline is available for both patients and referring physicians to schedule appointments within the following week. According to a written protocol, a secretary asks for the results of patients’ mammography and/or breast US reports (tumor size and imaging characteristics) to determine ahead of time whether the patient will need to be seen first by a surgeon or an oncologist. Then, the institution sends by regular mail a leaflet describing the outpatient breast clinic, the medical staff, as well as the types of procedures they might undergo. Consultations and all diagnostic procedures are located in a dedicated facility of the OSC and are performed during a single visit once a week by four different medical breast specialists; namely, an oncologist, a surgeon, a radiologist, and an interventional pathologist skilled in FNA procedures and interpretation of cytomorphology. Lastly, a nurse navigator is fundamental to coordinate the entire diagnostic workup, including patient assistance and psychological support. 

### 2.2. Patient Management in Our OSC

In our OSC, patients with breast lesions are evaluated by the triple test. This test entails a clinical breast assessment, a mammography and/or US evaluation (or re-evaluation if needed) to quantify the risk of malignancy of each lesion according to the American College of Radiology (ACR) Breast Imaging-Reporting and Data System (BI-RADS) lexicon. If a breast mass is detected, an FNA is carried out. In such cases, a topical anesthetic patch is applied to the region of interest before sampling.

### 2.3. FNA Sampling and Cytopathological Diagnosis

In our study, the FNA-based cytological diagnosis of breast lesions was made by three rotating cytopathologists either freehand for large or advanced tumors during the first 2 years then with the systematic help of a radiologist for US guidance (Siemens Acuson S2000 Helx Evolution; 18 Hz probe, München, Germany). Samples were collected with 2 to 3 multidirectional aspirates using a 10 mL syringe connected with a 23- or a 25-gauge (25 mm in length) needle depending on the distance of the mass from the areola. Overall, it took 10 to 20 s to collect the specimens. Samples were then immediately spread onto single slides and quickly air-dried. Next, they were stained with a Romanowsky-derived rapid staining (Diff Quik^®^) for ROSE. This enabled us to obtain additional FNA sampling when the first two specimens were diagnostically inadequate. To streamline the diagnostic process, we did not carry out any special or immunohistochemical staining or cell block preparations. The cytomorphological interpretation was conducted by an interventional pathologist capable of interpreting clinicoradiological information without double microscopic examination. Afterwards, an immediate cytopathological written report of the diagnosis was delivered to the clinician. The OSCAR procedure is the process conducted by an interventional pathologist performing FNA of the mass, rapid staining of the cytological sample, and immediate delivering of a definitive diagnostic report to the clinician. Cytopathologic diagnoses were then categorized at that time according to a 4-tiered classification system (see Discussion), i.e., malignant, suspicious, benign, and nondiagnostic. Such classification was based on the cellular yield and composition of 6 or more cell clusters or on the presence of more than 10 intact bipolar cells per 10 medium-power fields (×200). Malignancy was diagnosed when the smears contained either pure malignant cells or a largely predominant population of tumor cells admixed with few benign-looking glandular cells. A diagnosis of suspicious cytology was rendered in the following cases: when occasional cells of atypical features were intermingled with predominant populations of benign-looking cells; when smears showed naked nuclei in the background but with large clusters of cells with irregular boundaries; when paucicellular smears contained areas of mucinous or necrotic stroma; and when papillary formations with or without atypical cells were present. A diagnosis of benign cytology was made when the smears revealed normal, hyperplastic, or apocrine cells, and/or when fluid contained only cellular debris. In addition, a benign cytology consistent with fibroadenoma was suggested when numerous naked nuclei were intermingled with collagen strands and with anastomosing clusters of benign or apocrine cells; moreover, fat necrosis—consisting of numerous macrophages, multinucleated cells, and necrotic debris—was also associated with benign cytology. A nondiagnostic cytology report was made when smears contained no glandular cells or only debris or hemorrhage. Any discordance between the ACR BI-RADS classification and the specific cytopathological diagnosis was discussed on-site by the team, and CNB was performed immediately.

### 2.4. Proposal to the Patient

On the same day and depending on the results of the multidisciplinary discussion, patients were either discharged and invited to a clinicoradiological follow up or advised on primary treatment. In the latter case, treatment decision-making was immediately shared with the patients in a second medical consultation session. In particular, when the triple test was concordant with a benign cytological diagnosis, the surgeon or the oncologist reassured the patient and prescribed a bi-annual US and/or mammography check-up to evaluate the lesion at an 18-month US-FU. At that time, if the US showed no changes in the lesion size and the BI-RADS category remained the same, the lesion was definitely classified as benign. On the other hand, if the triple test was concordant with a malignant or suspicious cytologic diagnosis, depending on the tumor size and the clinical and US axilla status, patients met either with a surgeon or with an oncologist to discuss whether to undergo primary surgery or neoadjuvant medical treatment. If they opted for surgery, prognostic and predictive biomarkers (ER, PgR, and HER2) were evaluated either on the pre-operative CNB performed the same day or later after lumpectomy. Instead, if patients opted for primary medical treatment, immediate CNB was performed for histopathological grading and assessment of biomarkers. Same day CNB was also systematic in several other cases. One case was when FNA yielded nondiagnostic samples; a second case was when there was a discrepancy between the clinicoradiological and cytomorphological results. Typical examples of the latter cases were when a patient presented with a BI-RADS 5 mass with a benign cytology or with a BI-RADS 3 mass with a malignant cytology; a third case was in patients presenting with large tumors eligible for neoadjuvant medical therapy; and a final case was when patients harbored centrally located tumors or bifocal lesions—cases for which breast conservative treatment with sentinel lymph node evaluation was usually precluded. 

### 2.5. Histopathological Diagnosis

Histopathological examination without double reading was performed by senior breast pathologists either on formalin-fixed paraffin-embedded tissues obtained from US-guided CNB (14-Gauge) or on surgical specimens routinely cut and then stained with hematein and eosin. The 2012 World Health Organization Classification of Tumors was used then re-classified using the 2019 one [117] for the purpose of our study. In this regard, when breast carcinomas contained more than one histological feature, only the main one was considered. When interpreting CNB or surgical samples, histopathologists had already been informed about patients’ clinicoradiological characteristics and cytopathological diagnoses. The non-operated patients either had refused surgery or were inoperable because their general status was insufficient for surgical intervention. As mentioned above, CNB was performed immediately after FNA when indicated, whereas surgery usually occurred within the following 4–6 weeks.

### 2.6. Data Collection

Clinical characteristics, imaging findings, and cytomorphological and histopathological diagnoses were prospectively recorded in our centralized hospital database storage system. The histopathological results and final tumor status revealed at the 18-month US-FU were extracted from the hospital computerized registered medical record. When the data from the US-FU were not available, a questionnaire was posted to patients who sent it back after completing the dates, the types of radiological examinations, and their relative results performed in the last 18 months.

### 2.7. Statistical Analysis

Continuous variables were presented as mean (standard deviation); categorical variables were expressed as frequencies (percentages) and compared between groups with a Fisher’s exact test. Diagnostic accuracy was calculated by assuming that the multiple breast masses present in each patient were independent. As measures of diagnostic accuracy are based on the assignment of true and false negatives and positives, the cytomorphological categories of suspicious and malignant were combined and nondiagnostic cases were excluded from the calculations. From there, sensitivity (Se), specificity (Sp), negative predictive value (NPV), positive predictive value (PPV), negative likelihood ratio (LR−), positive likelihood ratio (LR+), accuracy and their corresponding 95% confidence intervals (CI) were calculated for histopathology cytomorphologic assessments, US-FU, and overall cytomorphologic assessment, which combined histopathology and US-FU assessments. The 95% CIs for Se, Sp, and accuracy were calculated based on Clopper–Pearson, NPV, and PPV; 95% CI were calculated using standard logit, and LR− and LR+ with 95% CIs were calculated using the log method [118]. LR+ for each of the 4 categories of cytomorphologic diagnoses (with their 95% CI) were also computed [119,120]. Statistical analyses were performed using the statistical software R, version 4.0.4 [121].

## 3. Results

### 3.1. Clinicoradiological Characteristics

Table 1 describes the clinicoradiological characteristics of 1740 patients harboring 1820 breast masses sampled by FNA. During the first two years, FNA samples of advanced and very large breast tumors were performed without US guidance. Then, all FNA samples were carried out under US guidance.

### 3.2. Study Flow Chart

Figure 1 illustrates the flow chart of the study. 

### 3.3. Cytomorphological Diagnoses

Table 2 shows the cytomorphological diagnoses of 1820 breast masses including the 191 with incomplete data.

In the group of 1629 breast masses, cytomorphological diagnoses were verified by histology in 1138 (69.8%) cases and by US-FU in 491 (30.1%) cases. Of the 1629 masses, 50.1% (816 cases) were cytologically malignant and concordant with histology in 98.4% (803/816) and with US-FU in 1.5% (12/816) of the cases, reaching an overall concordance rate of 99.9% (815/816); 9.3% (152 cases) were cytologically suspicious and histologically malignant in 80.2% (122/152 cases); 37.5% (611 cases) were cytologically benign and concordant with histology in 18.8% (115 cases) and with US-FU in 77.2% (472 cases), reaching an overall concordance rate of 96.1% (587/611); 3.1% (50 cases) were nondiagnostic, 18 of which (36%) turned out to be malignant after histological examination. False-negative and false-positive cytomorphological diagnoses were 1.5% (25/1629) and 0.06% (1/1629), respectively.

In the group of 177 patients with missing data, 3.1% (26/842) were malignant, 1.3% (2/154) were suspicious, 2.1% (160/771) were benign, and 5.7% (3/53) were nondiagnostic specimens for cytomorphological evaluation. No significant differences were found between the groups of 1629 and 1820 masses.

Table 3 displays the main clinicoradiological and histopathological characteristics of 228 breast masses histologically verified (the group with the missing data being excluded). Cytomorphology was suspicious in 152 cases, unsatisfactory in 50 cases, false negative in 25 cases, and false positive in 1 case. The mean age of patients was similar in the first three groups (59 years; range: 57–61). The mean size of breast masses was significantly smaller in the group of nondiagnostic specimens even though 96% (48/50) of FNA specimens were performed under US guidance. Similarly, 72% (18/25) of FNA samples were performed under US guidance in the group of false negatives. In both nondiagnostic and false-negative groups, breast masses were mainly classified as BI-RADS of intermediate (62% of BI-RADS 4; 31/50) and high (96% of BI-RADS 5; 24/25) risk.

In the group of patients with suspicious cytomorphology (n = 152), the mean age was 57 years (+/− 13.4), the mean size of breast masses was 17.1 mm (+/− 15), and US-FNA was performed in 61.8% (94/152) of the cases. It comprised a first subgroup of 30 suspicious cases with either benign histology (n = 29) or with a (n = 1) reassuring 18-month US-FU. Sampling was performed under US guidance in 77% (23/30) of cases, and the results were categorized as BI-RADS 4 and BI-RADS 5 in 63.3% (19/30) and 20% (6/30) of the cases, respectively. Interestingly, the most frequent histological diagnoses were benign fibroadenomas (n = 9); papilloma (n = 9); adenosis (n = 3); and radial scar (n = 2). In the second subgroup of 122 suspicious cases with malignant histology, FNA was carried out under US guidance in only 59.8% (73/122) cases. In these latter cases, the rates of BI-RADS 4 and BI-RADS 5 were 33.6% (41/122) and 57.4% (70/122), respectively. Histopathology of the 122 cases showed 3 cases of pure ductal carcinoma in situ (DCIS) of low (n = 1) and high (n = 2) nuclear grade, 99 cases of invasive ductal carcinoma (IDC) of grade I (n = 46), II (n = 31), III (n = 15), or unknown (n = 7), 18 cases of grade I invasive lobular carcinoma (ILC), 1 case of malignant myoepithelioma, and 1case of a phyllode tumor of intermediate grade.

In the group of patients with nondiagnostic FNA specimens, histology showed 13 IDC of grade I (6 cases including 1 tubular carcinoma) or grade II (7 cases including 1 apocrine carcinoma), and 5 cases of ILC grade I.

In the group of patients with false-negative cytomorphology, histology showed 2 cases of pure DCIS of intermediate nuclear grade, 17 cases of grade I (including 5 tubular carcinomas) IDC and 3 IDC of grade II (including 1 mucinous adenocarcinoma), and 6 cases of ILC (1 grade I and 5 grade II).

The patient with a cytologically false-positive case corresponded to a 49-year-old woman presenting with a 20 mm BIRADS 4 left breast mass, which was sampled by free-hand FNA but then corrected by evaluation of the CNB.

### 3.4. Diagnostic Accuracy Measures

Table 4 displays the different measures of diagnostic accuracy with corresponding 95% CIs for condensed cytomorphologic diagnosis with final tumor status in terms of sensitivity (Se), specificity (Sp), positive predictive value (PPV), negative predictive value (NPV), positive likelihood ratio (LR+), negative predictive ratio (LR−), and accuracy. As already mentioned (see Material and Methods), nondiagnostic specimens were excluded from analysis and suspicious diagnoses were grouped with malignant. From a total of 1579 cytomorphological diagnoses, 1094 and 485 were, respectively, verified by histology or US-FU. While sensitivity and specificity cannot be used on their own to estimate an individual’s probability of disease, likelihood ratios (LRs) provide a practical way of interpreting test results that can be translated to individual patients [123]. Table 4 illustrates positive likelihood ratios (LRs+) with 95% CIs according to the different categories of cytomorphology in the group of 1629 breast masses verified by either histology or an 18-month US-FU. In addition, disease prevalence within a population impacts predictive values. To avoid having to convert pre- and post-test probability to pre- and post-test odds, a Fagan’s nomogram [124,125] shows the post-test likelihood of malignancy as a function of pre-test probability (prevalence) of the OSCAR procedure in our series is illustrated in Figure 2.

## 4. Discussion

Our results underline the high diagnostic performance of the entire OSCAR procedure in patients with breast masses sampled by FNA and immediately interpreted by cytomorphology in the context of a multidisciplinary OSC setting. In addition, by facilitating consultations between the multidisciplinary team and patients, OSCAR enables clinicians to deliver definite cytopathological diagnosis to the patient within a single day, thereby streamlining the whole process of treatment decision making and reducing the levels of anxiety associated with long diagnostic waiting times. Furthermore, we showed that the OSCAR procedure can also reliably be considered as a rule in testing and identify patients with breast masses requiring CNB, thereby sparing many women the physical and emotional distress of unnecessary biopsies.

Diagnostic mammography and US assessments were categorized on the basis of the ACR BI-RADS lexicon which efficiently classifies the risk of malignancy in patients with breast diseases. Our study used the fourth edition of the system, which was published in 2003 [126]. In our series of 965 histologically malignant cases, 23% and 74% were classified as BI-RADS 4 and BI-RADS 5, respectively. Later versions of the system added mammographic aspects, measures of likelihood of malignancy, and management recommendations and subclassified BI-RADS 4 into BI-RADS 4A, 4B, and 4C categories [126,127]. However, although BI-RADS 4C and 5 breast lesions are associated with high risk of malignancy, they may also correspond to benign lesions [128].

Performing FNA under US guidance is also known to be an important factor for enhancing diagnostic accuracy [63]. Not surprisingly, our study highlights that from 2004 to 2007, FNA under US guidance increased from 56.4% to 69%. This phenomenon was further confirmed in a 2016 study highlighting its increased popularity in tandem with the increased detection of small-sized breast masses thanks to national screening and more advanced imaging technologies [114].

Our study lends support to earlier literature indicating the diagnostic efficiency of US-FNA for cytological diagnosis of breast masses. We confirm that achieving diagnostic accuracy is tightly dependent on the presence of an interventional pathologist. Indeed, our study has demonstrated that having a pathologist with skills in performing and interpreting FNA specimens can ensure a diagnostic accuracy equivalent to the quality assurance thresholds recommended for small tissue specimens (FNAC and CNB) by several international guidelines. Among these are the “*Guidelines for cytology procedure and reporting on fine needle aspirates*” [129,130] with measurements of likelihood ratios [131]; “*European guidelines for quality assurance in breast cancer screening*” (section 5.5.3: Cytology/histology quality assurance) [132] published in 2006; and “*Guidelines for non-operative diagnostic procedures and reporting in breast cancer screening*” (2nd edition; document G150) [133] published in August 2021. Some authors have suggested that double reading selected cases by an expert cytopathologist might also be essential to improve cytodiagnostics and the quality of patient care [134]. However, we speculate that although this may be a useful tool at the beginning of a diagnostic learning curve, it might then slow down the entire diagnostic work-up.

Importantly, the OSCAR procedure resulted in low rates of nondiagnostic FNA specimens (3.1%), false negatives (1.5%), and false positives (0.06%). Factors influencing nondiagnostic FNA specimens include training, experience, and expertise in aspiration techniques [135,136,137,138,139,140,141,142], nature, size, and histotype of the lesion, as well as the number of passes [91,143,144,145,146,147,148]. Moreover, a large systematic review and meta-analysis synthesizing evidence across different anatomic locations indicates that the application of ROSE, or lack thereof, is another critical factor influencing FNA diagnostic outcomes [111]. Importantly, lowering the rates of nondiagnostic preoperative FNA specimens not only guarantees fast and accurate diagnoses but also decreases the costs of the diagnostic workup [149,150] and increase its cost-effectiveness [151]. Accordingly, over the last decade, much effort has been dedicated to developing new guidelines to reduce the unsatisfactory/non diagnostic category rates of FNA samples [152].

Regarding the factors contributing to the occurrence of false-negative and false-positive FNA cases, studies have shown that morphological difficulties associated with low-grade malignancies may play a key role. This was illustrated in our series of 25 cases of false-negative cases of breast masses comprising 14 IDC grade I (with 5 tubular carcinomas) and 6 cases of ILC (1 grade I and 5 grade II). Of note, only one case was classified as low risk (BI-RADS 3) by imaging. The BI-RADS 4 false-positive FNA case was fortunately further explored by CNB, and the patient was treated by conservative surgery. These false-negative and false-positive cases underline the critical role of the triple test. Discordance between the specimens and the imaging features led us to verify all the cases histologically. Indeed, of all the 152 cases diagnosed as suspicious by FNAC, 30 (19.7%) resulted as benign after histological examination (n = 29) or at the 18-month US-FU (n = 1). Moreover, whereas 5 cases (16.7%) were BI-RADS 3 and 19 cases (63.3%) were BI-RADS 4, 6 (20%) cases were BI-RADS 5. Of the 122 cases diagnosed as suspicious by cytomorphology (80.3%) and malignant by histology, 70 cases (57%) were BI-RADS 3, 41 cases (33.6%) were BI-RADS 4, and 7 cases (5.7%) were BI-RADS 5. Histologically, 3 were pure DCIS of low and high nuclear grade and 99 were IDC including 17 lobular, 11 tubular, 4 mucinous, 2 papillary, 3 metaplastic, 1 cribriform, and 1 apocrine type of carcinoma. These findings suggest that since most of these histotypes are associated with cytological difficulties [60,153,154], they should be analyzed histologically after cytologic evaluation.

We used a 4-tiered system for reporting breast FNA cytomorphology instead of the 5-tiered system recommended by the National [129], European [132], American [155], and international organizations [156,157]. Most organizations recommend the additional use of an “atypical” category because it maintains a high predictive value of cytological malignancy while retaining a high degree of diagnostic sensitivity. However, at the time of our study (2004–2007), the clinicians at our institution were reluctant to use this category, mainly because it could create confusion both for patients and for management. Nowadays, however, this category is progressively being incorporated in our routine practice. Despite the introduction of the atypical category, the interpretation of certain cytomorphological findings still remains challenging. Some problems of interpretation may also arise in some unclear cases of breast CNB specimens, particularly those categorized as uncertain malignant potential (B3) and atypical or undetermined (B4). Owing to the heterogeneous and complex nature of these lesions, there is always a risk of under- or overestimating the occurrence of ductal carcinoma in situ of low and intermediate nuclear grade, as well as fibroepithelial and papillary neoplasms [158,159,160,161,162,163,164,165,166,167,168,169,170]. The diagnosis of cytological atypia may be challenging, and the morphological features of breast masses sampled by FNA or CNB do not always overlap. Accordingly, several studies have suggested using both approaches as sequential [164] and complementary methods [171,172,173,174,175,176]. Future prospective studies based on the recent *International Academy of Cytology (IAC) Yokohama System for Reporting Breast Fine Needle Aspiration Biopsy Cytopathology* providing a more detailed definition, along with illustrations, of cytologic atypia [156,157,177] are needed. It is also important to mention that management options have to take into account the considerable differences in practice between well-resourced and middle- or low-income countries, as detailed in the book explaining and illustrating the IAC Yokohama system [157].

A unique and fundamental aspect of the OSCAR procedure is obviously the presence of an experienced and skilled interventional pathologist [136,178,179,180,181] capable of both performing FNA preferentially under US guidance [182,183,184,185,186] and delivering a swift diagnosis using rapid stains [187,188,189,190,191,192,193]. On the other hand, some ROSE-based models performed by nonpathologists have been described. In these cases, a remote diagnosis is carried out by a pathologist using telecytology [194]. Although the latter method is cost-effective and, therefore, applicable in low-income settings, the presence of a pathologist is indisputably critical. Having a pathologist on-site gives the possibility to immediately discuss any type of difficulties, including discordance between clinical examination and/or imaging interpretation. Furthermore, having an in-house pathologist also gives the pathologist herself/himself the opportunity to interact with the whole clinicoradiological team in person and to participate in a “mini on-site” tumor board, if needed. This procedure is tremendously advantageous as it streamlines the entire diagnostic pathway, benefitting the overall care of patients.

Long waiting times for pre-operative assessments are physically and emotionally detrimental for patients. In conventional clinical settings, the excruciating waiting times needed to fix an appointment with a breast specialist, to have image-guided sampling, if needed, to receive a morphological diagnosis based on FNA and/or CNB, and to schedule further appointments to decide on a course of treatment are major sources of discomfort and anxiety for patients. The failure of the 2-week wait rule introduced in the UK [195] by the Department of Health (Health Service Circular HCS 1998/242) prompted the development of rapid access breast specialist clinics [116,196,197,198] along with their corresponding recommended quality indicators [199]. In this context, the overarching goal of a breast OSC has been to reduce the waiting times of the entire process of the pre-operative assessment while maintaining a high level of diagnostic performance and minimal patient anxiety. This has been made possible by having a dedicated team of breast cancer specialists work in tandem to provide each patient with the best possible care in a minimal period of time. In our experience, we have seen that the logistics of such patient-centered organization is relatively sophisticated, requiring highly professional medical secretaries able to handle hotlines and nurse navigators able to assist patients throughout the entire course of the diagnostic pathway [114].

Although FNA may significantly improve the diagnostic pathway of patients with breast masses, its use remains a matter of debate. Indeed, the preference for CNB over FNA for breast lesions is still clearly stated in both the 2016 edition of the *NHS Breast Screening Programme* in the UK [200] (4th edition, document number 49) and in the updated guidelines for non-operative diagnostic procedures in breast cancer screening published by the Royal College of Pathologists [201] (document number G150, version number 2) in August 2021. FNA is only recommended for lymph node assessment. Yet, other organizations acknowledge the useful implementation of breast FNA, provided that skilled operators perform the procedure and experienced pathologists interpret the cytomorphologic features of FNA samples. Among these institutions are the NCCN [202], ESMO [203], EUSOMA [199,204], and the Japanese Breast Cancer Society [205]. In addition to reducing delays in diagnosis while ensuring diagnostic accuracy, some studies have suggested that OSCs might also contribute to easing the levels of patients’ anxiety while waiting for diagnosis [206,207]. The question of whether speedier diagnosis can actually lower women’s psychological distress and improve the overall care is still a matter of debate [208,209]. However, in our experience, most of our patients (80%) expressed high levels of satisfaction with the care we provided. Only a limited number of patients experienced anxiety, which was most likely associated with difficulties in establishing a good-quality relationship with their therapist [210].

Our study did have some limitations. The most obvious limitation was the single center, observational and descriptive nature of our series [211,212]. Despite being prospectively registered in the central informatics system of our institution, its observational and retrospective aspects may have given rise to some bias and confounding factors [213,214,215,216]. Another limitation was that the cytomorphological diagnoses were reported according to a 4-tiered and not a 5-tiered classification system, as it is now recommended. A third and final limitation was that the study was not randomized since a subset of patients underwent only long-term (18-month) US-FU without histological assessment. The rationale behind the choice of a non-randomized study [217] was to avoid ethical issues commonly associated with blinded studies.

Despite these limitations, our study has several strengths. These strengths included the large consecutive series of patients analyzed and its reporting, which was conducted according to published guidelines and specifications [122,218,219,220,221,222]. Another strength was that it was focused on patients with breast masses identified by mammography and US. By doing so, we eliminated confounding factors such as pure microcalcifications, architectural distortion, and asymmetries of breast imaging.

The main advantages of the OSCAR procedure are the possibility of triaging patients using a minimally invasive, rapid, safe, and low-cost technique identifying patients who are the best candidates for core needle biopsy. In addition, it significantly reduces the turn-around time for benign breast masses and therefore patients’ anxiety. On the other hand, this patient-centered approach needs a multidisciplinary oncoteam, and its main disadvantage is the need for an interventional pathologist able to deliver an immediate diagnosis based on cytology. Furthermore, this approach can only be performed in patients with breast masses.

Our findings indicate that the OSCAR procedure makes a very efficient approach for reaching a definitive diagnosis within a single day. Thanks to this procedure, we were also able to distinguish between women with malignant cytology who actually required CNB from those with benign cytology who were reassured and spared the physical and emotional distress of unnecessary CNB. We also demonstrated that the OSCAR procedure can be used as a rule-in test, as indicated by the main diagnostic indicators of performance. Similarly, we showed that it may also be used as a rule-out test. Indeed, at least in our study, we evidenced that the combination of clinical and radiological findings may facilitate the detection of discordance, a main indicator of CNB. Implementing the OSCAR procedure in a multidisciplinary OSC for breast masses enables clinicians to triage patients [223,224] adding a layer of confidence to the imaging features for patients requiring CNB. Decreasing the number of unnecessary CNB also has an important financial impact on the health care system [114]. The ideal setting to apply OSCAR is a breast referral/specialist clinic with a patient-centered pathway, a multidisciplinary team, and a tumor board, as recommended in the EUSOMA [225,226,227,228] and EUSOBI [204,229] guidelines. Undeniably, our internal results need to be verified prospectively and validated externally in other settings [230] where the prevalence of breast cancer may differ significantly [231]. Finally, our data highlight time-saving aspects of our FNA-based cytomorphologic approach compared to CNB and suggest that the OSCAR procedure may be recommended as a triage test for indicating CNB in high-income countries and as an alternative approach to conventional diagnostic pathways in middle- or low-resource countries where the health system cannot afford the costs of CNB [232,233,234,235,236] and infrastructures of histopathology laboratories [237,238].

## 5. Conclusions

Our study clearly shows that the OSCAR procedure is a highly reliable diagnostic approach and a perfect test to select patients requiring CNB when performed by interventional cytopathologists in a multidisciplinary setting. Additionally, it drastically reduces the percentage of nondiagnostic specimens and diagnostic turn-around time. Furthermore, it is an efficient and powerful first-line diagnostic approach for patient-centered care.

## Figures and Tables

**Figure 1 cancers-15-04967-f001:**
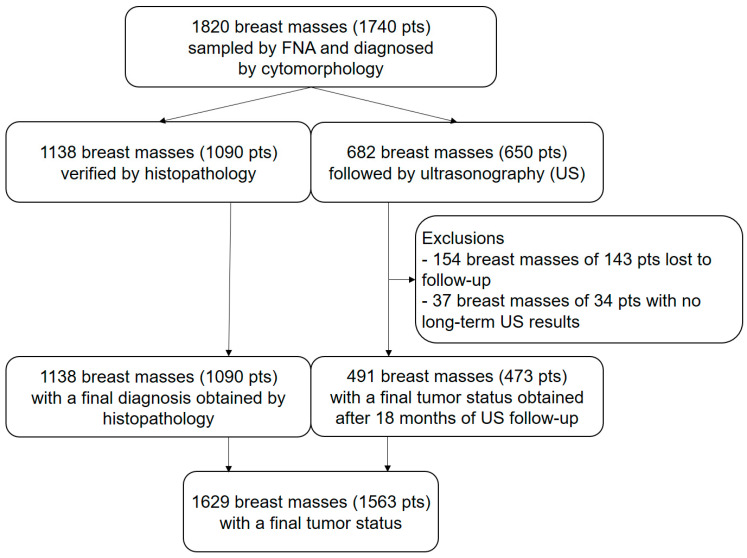
Illustrates the study flow chart according to the STARD guidelines [122]. From an initial set of 1820 breast masses (1740 pts), US-FU data were lacking in 177 patients with 191 (10.5%) masses. Of the remaining 1629 breast masses, 1138 (62.5%) were verified by histopathology and 528 (29%) were evaluated at the 18-month US-FU. Final tumor status was therefore available for 1629 (89.5%) masses (1563 patients).

**Figure 2 cancers-15-04967-f002:**
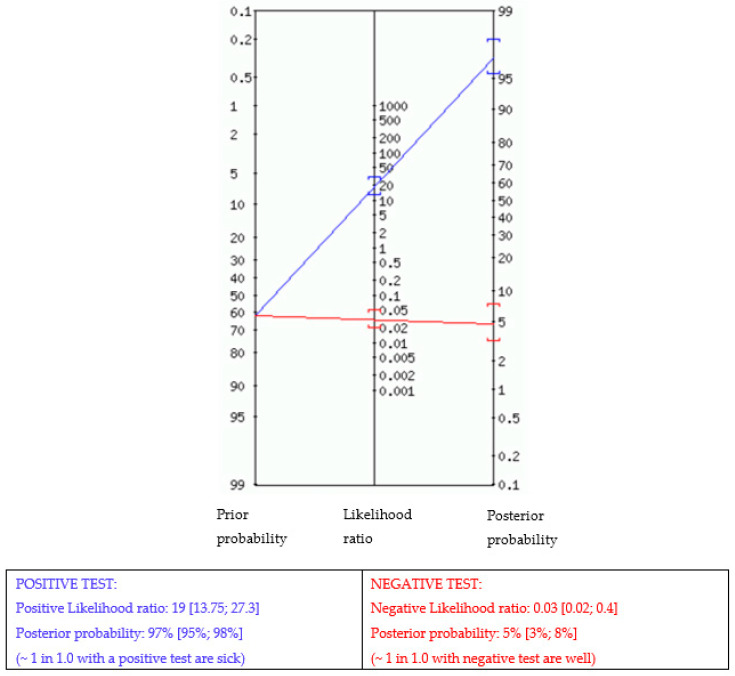
Fagan nomogram. The nomogram displays the probability that a patient has the disease after a positive or negative test. Prior probability (odds): 62%.

**Table 1 cancers-15-04967-t001:** Clinicoradiologic characteristics of the 1740 patients with 1820 breast masses.

**Characteristic ^a^**	
**Age (years)**	56 ± 14
**Number of lesions ^b^**	
1 2 3	1661 (95.4) 78 (4.5) 1 (0.1)
**Size (mm)**	21 ± 30
**BI-RADS ^c^**	
1 2 3 4 5 Unknown	5 (0.3) 103 (5.7) 472 (25.9) 453 (24.9) 777 (42.7) 10 (0.5)
**US-FNA ^b^**	1114 (65.5)

^a^ Mean ± sd or n (%); ^b^ Per patient (n = 1740); ^c^ According to the American College of Radiology (ACR) classification (n = 1820).

**Table 2 cancers-15-04967-t002:** Cytomorphologic diagnoses in 1820 breast masses and comparison with the group of 1629 masses with final tumor status obtained either by histopathology (Histo) or by the 18-month ultrasonography follow-up (US-FU).

	Final Tumor Status						
	Malignant		Benign				
Cytomorphology	Histo	US-FU	Histo	US-FU	Total	NA *	Total
**Malignant**	803	12	1	0	816	26	842
**Suspicious**	122	0	29	1	152	2	154
**Benign**	24	1	115	471	611	160	771
**Nondiagnostic**	18	0	26	6	50	3	53
Total	967	13	171	478	1629	191	1820

* NA = breast masses corresponding to patients lost to follow-up (n = 143) or without (n = 34) an 18-month US-FU.

**Table 3 cancers-15-04967-t003:** Clinicoradiologic and histopathologic characteristics of 288 breast masses.

Characteristics	Suspicious (n = 152)	Nondiagnostic (n = 50)	False Negative (n = 25)	False Positive (n = 1)
Mean age, years (+/− sd)	57 (13.4)	61 (9)	60 (10)	49
Mean size, mm (+/− sd)	17.1 (15.0)	10.62 (6.3)	16.12 (14.19)	20
US-guided FNAC, n (%)	94 (61.8)	48 (96)	18 (72)	0 (0)
BI-RADS, n	152	50	18	1
2	0	2	0	0
3	12	7	1	0
4	60	24	10	1
5	76	17	14	0
Unknown	4	0	0	0
Final tumor status, n	152	50	25	1
Benign: n	30	32	0	
Follow-up	1	6	0	
Histologically verified	29	26	0	^j^ 1
Malignant, n	122	18	25	
Ductal carcinoma in situ	^a^ 3	0	^g^ 2	
Invasive ductal ca	99	13	17	
Grade I	^b^ 46	^e^ 6	^h^ 14	
Grade II	^c^ 31	^f^ 7	^i^ 3	
Grade III	^d^ 15	0	0	
Unknown	7	0	0	
Invasive lobular ca	18	5	6	
Grade I	18	0	1	
Grade II	0	5	5	
Grade III	0	0	0	
Phyllode tumor (intermediate grade)	1	0	0	
Malignant myoepithelioma	1	0	0	

n = number; ^a^: Including 3 ductal carcinomas in situ (DCIS) not otherwise specified (NOS) of low (n = 1) and high (n = 2) nuclear grade. ^b^: Including 12 invasive duct carcinomas (IDC) NOS associated with DCIS NOS of low (n = 7) and high (n = 5) nuclear grade, 8 tubular, 2 mucinous, 1 papillary, 1 cribriform, and 1 apocrine invasive carcinoma. ^c^: Including 6 IDC NOS with DCIS of intermediate nuclear grade, 3 tubular, 1 mucinous, and 1 papillary carcinoma. ^d^: Including 3 metaplastic, 1 mucinous, and 1 IDC NOS associated with 1 DCIS NOS of high nuclear grade. ^e^: Including 2 tubular and 1 IDC NOS associated with invasive lobular carcinoma grade I. ^f^: Including 1 apocrine and 1 IDC NOS associated with DCIS of intermediate nuclear grade. ^g^: Corresponding to 2 DCIS of intermediate nuclear grade. ^h^: Including 5 tubular and 3 IDC NOS associated with DCIS of low (n = 2) and high (n = 1) nuclear grade. ^i^: Including 1 mucinous adenocarcinoma and 1 IDC NOS associated with invasive lobular carcinoma grade I. ^j^: Fibroadenoma with florid benign ductal hyperplasia.

**Table 4 cancers-15-04967-t004:** Measures of diagnostic accuracy and corresponding confidence intervals for condensed cytomorphologic diagnosis with final tumor status.

		Cytomorphology Overall (n = 1579)	Cytomorphology–Histopathology Confirmed (n = 1094)	Cytomorphology–Ultrasound Follow-Up (n = 485)
Se	value	97.40%	97.47%	92.31%
	95% CI (LL-UL)	96.19–98.31%	96.26–98.37%	63.97–99.81%
Sp	value	94.98%	79.31%	99.79%
	95% CI (LL-UL)	92.94–96.56%	71.80–85.58%	98.83–99.99%
PPV	value	96.80%	96.86%	92.31%
	95% CI (LL-UL)	95.48–97.81%	95.55–97.87%	63.97–99.81%
NPV	value	95.91%	82.73%	99.79%
	95% CI (LL-UL)	94.02–97.33%	75.41–88.61%	98.83–99.99%
LR+	value	19.39	4.71	435.69
	95% CI (LL-UL)	13.75–27.32	3.42–6.48	61.11–3106.14
LR−	value	0.03	0.03	0.08
	95% CI (LL-UL)	0.02–0.04	0.02–0.05	0.01–0.51
Accuracy	value	96.45%	95.06%	99.59%
	95% CI (LL-UL)	95.42–97.31%	93.61–96.27%	98.52–99.95%

Se = sensitivity; Sp = specificity; PPV = positive predictive value; NPV = negative predictive value; LR+ =positive likelihood ratio; LR− = negative likelihood ratio; CI = confidence interval; LL = lower limit; and UL = upper limit. NOTE: 95% CI for Se, Sp, and accuracy are Clopper–Pearson CIs; 95% CI for PPV and NPV are standard logit CIs; and 95% CI for LR+ and LR− use log method.

## Data Availability

The data presented in this study are available in this article.

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
