# Peer review of "Real-World Diagnostic Accuracy of the On-Site Cytopathology Advance Report (OSCAR) Procedure Performed in a Multidisciplinary One-Stop Breast Clinic"

_cancers, 2023, doi:10.3390/cancers15204967_

Round 1
Reviewer 1 Report
The present retrospective research paper entitled “Real-world diagnostic accuracy of the on-site cytopathology advance Report (OSCAR) procedure performed in a multidisciplinary one stop breast clinic” by Suciu et al. is novel and very well written. The authors cover almost all aspects. I have the following comments that are needed to be addressed.
Comment 1. What are the factors needed to be considered before the procedure? Kindly explain separately.
Comment 2. What are the limitations of the OSCAR procedure? Kindly explain.
Comment 3. Kindly write the conclusion separately without reference.
Author Response
Answers to Reviewer 1 comments
Comment 1. What are the factors needed to be considered before the procedure? Kindly explain separately.
The requirements to be considered before the procedure are detailed in Patients and Methods/Organization of the breast one stop clinic/second paragraph: lines 113-125.
Comment 2. What are the limitations of the OSCAR procedure? Kindly explain.
We have added an explanation in the text (lines 504-511) accordingly.
Comment 3. Kindly write the conclusion separately without reference.
We have included a concise conclusion separately without any references: lines 536-542.
Reviewer 2 Report
The study titled "Diagnostic Accuracy of the On-Site Cyto-2 Pathology Advance Report (OSCAR) Procedure Performed in a Multidisciplinary One-Stop Breast Clinic" has been evaluated. The authors of this study performed histopathological verification or an 18-month US evaluation when a benign cytology was consistent with the components of the triple test. In total, histology was available for 1138 masses, while 491 masses were analyzed at the 18-month US-FU. FNA specimens were morphologically nondiagnostic in 3.1%, with false negatives observed in 1.5% and only one false-positive (0.06%). The breast cancer prevalence was 62%. The diagnostic accuracy measures of the OSCAR procedure with their 95% confidence intervals (95% CI) were as follows: sensitivity (Se) = 97.4% (96.19-98.31); specificity (Sp) = 94.98% (92.94-96.56); positive predictive value (PPV) = 96.80% (95.48-97.81); negative predictive value (NPV) = 95.91% (94.02-97.33); positive likelihood ratio (LR+) = 19.39 (13.75-27.32); negative predictive ratio (LR-) = 0.03 (0.02-0.04), and; accuracy = 96.45% (95.42-97.31). The respective positive likelihood ratio (LR+) for each of the four categories of cytopathological diagnoses (with their 95% CI) that are malignant, suspicious, benign, and nondiagnostic were: 540 (76-3827); 2.69 (1.8-3.96); 0.03 (0.02-0.04), and; 0.37 (0.2-50 0.66). The study indicates that the OSCAR procedure is a highly reliable diagnostic approach and a perfect test to select patients requiring core needle biopsy (CNB) when performed by interventional cytopathologists in a multidisciplinary and integrated OSC setting. In addition to drastically limiting the rate of nondiagnostic specimens and diagnostic turn-around time, OSCAR is an efficient and powerful first-line diagnostic approach for patient-centered care.
I have reviewed the manuscript and have some major concerns and highlighted some of them in the manuscript file. The number of references for the research article should be between 50-75. Additionally, the authors should add the pros and cons of the study.
Moderate editing of English language required
Author Response
Answers to Reviewer 2 comments
The number of references for the research article should be between 50-75.
According to the Instructions for authors : https://www.mdpi.com/journal/cancers/instructions#references
there is no restriction regarding the number of references.
We think that all included references [1-105] are important as they reflect the evolution of fine needle aspiration cytology of the breast over the years and emphasize the emerging recent and crucial role of interventional cytology.
Additionally, the authors should add the pros and cons of the study.
The limitations and strengths of our study are detailed in the discussion lines 488-497 and 498-503, respectively.
We have added a paragraph summarizing the main advantages and disadvantages of the OSCAR procedure (lines 504-511) accordingly.
Major concerns highlighted in the text.
Modifications have been included in the text when applicable.
The subtitle should re organized. like "organization barest OSC data base"
The subtitle has been modified in the text accordingly.
The font size should be readable and enlarged. Remove Bold font and enlarge the font size
The font size has been enlarged and Bold font removed accordingly.
Improve figure resolution
The figure resolution has been improved accordingly.